# Respiratory Infection and Inflammation in Cystic Fibrosis: A Dynamic Interplay among the Host, Microbes, and Environment for the Ages

**DOI:** 10.3390/ijms24044052

**Published:** 2023-02-17

**Authors:** Christiaan Yu, Tom Kotsimbos

**Affiliations:** 1Department of Respiratory Medicine, Alfred Health, Melbourne, VIC 3004, Australia; 2Department of Medicine, Monash University, Alfred Campus, Melbourne, VIC 3004, Australia

**Keywords:** cystic fibrosis, CTFR modulators, environment, host, infection, inflammation, immunity, microbes

## Abstract

The interplay between airway inflammation and infection is now recognized as a major factor in the pathobiology in cystic fibrosis (CF). A proinflammatory environment is seen throughout the CF airway resulting in classic marked and enduring neutrophilic infiltrations, irreversibly damaging the lung. Although this is seen to occur early, independent of infection, respiratory microbes arising at different timepoints in life and the world environment perpetuate this hyperinflammatory state. Several selective pressures have allowed for the CF gene to persist until today despite an early mortality. Comprehensive care systems, which have been a cornerstone of therapy for the past few decades, are now revolutionized by CF transmembrane conductance regulator (CTFR) modulators. The effects of these small-molecule agents cannot be overstated and can be seen as early as in utero. For an understanding of the future, this review looks into CF studies spanning the historical and present period.

## 1. A Brief History

Since the first description of cystic fibrosis (CF) in 1938, airway infections have been recognized to play an important role in CF-associated disease. The earliest postmortem studies in infants demonstrated evidence of bronchiectasis, bronchitis, lobar pneumonia, and the presence of *Staphylococcus aureus* [1]. During a heat wave in 1948, New York City saw a high proportion of children with CF presenting with dehydration from excessive salt loss in sweat. This led to the development of the sweat chloride test, which is a cornerstone for the diagnosis for CF [2]. Utilizing quantitative pilocarpine iontophoresis, this test detects the inability of the CF sweat gland to reabsorb sodium chloride. The high clinical utility of this test in turn led to the assumptive leap that dehydrated, dysfunctional mucociliary host defense was the main initial driver for increased respiratory infections and subsequent chronic airway inflammation [Figure 1], which would be punctuated with acute exacerbations over time, leading to varying stepped trajectories of progressive lung function decline toward end-stage lung disease. Decades of work trying to understand the molecular basis of this disorder culminated in a major breakthrough in 1989 when the location of the CF gene on chromosome 7 and mutation F508Del was identified [3]. At the time of this discovery, the life expectancy of patients with this disease was approximately 20 years [4]. Further efforts by the CF clinical, scientific, and advocacy community in the subsequent years led to further significant advances resulting in dramatic cumulative improvements in survival. These advances included a more complete understanding of the importance of adequate salt/water and nutrition replacement, the evolution of multiple airway clearance and antibiotic treatment strategies, and most recently, the innovative development of specific interventions to improve the functioning of the abnormal CF transmembrane conductance regulator (CFTR) protein, which is at the core of the CF phenotype [5,6,7,8,9].

The CFTR protein plays an integral part in the regulation of salt and water in a diverse range of epithelial cells, including the airways, gastrointestinal and reproductive tracts, and sweat glands. This function is lost or diminished in CF, resulting in dysregulated chloride and sodium movement across the cell membrane that is particularly inappropriate for specific local environments where the concentration of these ions is very important. In various organ systems, therefore, exocrine secretions are no longer hydrated and become thick and vicious. The first manifestation of CF often occurs in the gastrointestinal tract in the form of meconium ileus. The obstruction of the bowel due to a thick meconium occurs in 20% of CF patients. Correspondingly, 98% of infants with meconium ileus end up being diagnosed with CF [7]. The subsequent years of childhood, adolescence, and adulthood feature pulmonary exacerbations [10]. These recurrent episodes of acute respiratory symptoms and unscheduled hospital visits are commonly triggered by infections and perpetuated by exaggerated inflammation, leading to irreversible lung damage. In the 21st century, in most developed countries, the mean life expectancy in CF patients has doubled to over 40 years old [6,11,12,13]. In spite of these developments, CF remains the most common life-shortening autosomal recessive genetic disorder among Caucasians [14]. Above all, pulmonary disease continues to be the prime cause of morbidity and mortality in the CF population [15].

The recent development and now widespread use of CFTR modulator therapies for various class mutations have single-handedly revolutionized current CF care treatment strategies for 90–95% of the Australian CF population of over 3600 patients eligible for these agents (notwithstanding current accessibility barriers in many parts of the world) [12]. Either by potentiating the channel-open probability of CTFR protein or increasing the quantity of functional CFTR delivered to the cell surface, these agents have the potential to improve and stabilize lung function and dramatically lower pulmonary exacerbation rates including those precipitated by proven respiratory viruses [16,17,18]. For the first time in CF history, we have a specific pharmacological intervention tackling the upstream CFTR defect. Not only will these modulators hold the key to new treatment paradigms for CF, but specific patterns of response heterogeneity in different CF sub-cohorts (based on CF mutation abnormality, age, pre-existing disease, comorbidities, and various environmental and modifier gene confounders) will also give us the opportunity to much better understand the underlying pathobiology (mutation classes are discussed in depth in Section 3).

## 2. The Host and the Environment: Inflammation and Infection (and More Inflammation)

At birth, the lungs and submucosal glands are histologically normal. Soon after, however, an extensive neutrophilic response in the peribronchial areas is often identified. A robust inflammatory response is, therefore, not uncommon in toddlers, even in culture-negative patients, which in turn is usually associated with mucopurulent plugging of medium-sized bronchioles [19]. With increasing age, elevated IL-8 and neutrophil elastase dominate the airway micro-environment, leading to local chondrolysis and proteolysis of airway support tissue [20]. Over the long term, structural damage and bronchiectasis are the pathophysiological endpoints of this chronic injury that is often further complicated by episodes of acute exacerbation. Although the CF airway environment most likely reflects an early excessive inflammatory response, this is very quickly confounded by episodes of acute viral respiratory infection in childhood, as well as colonizing bacterial organisms, making the interaction between all these factors somewhat difficult to delineate. With this in mind, the unchecked influx of neutrophils, soluble inflammatory mediators such as tumor necrosis factor alpha, IL-1, and leukotriene B, and upregulated innate and adaptive immune system factors generally create multiple auto-amplifying loops of inflammation that are not only difficult to switch off but also promote airway microbial colonization and chronic infection [21]. Indeed, each acute exacerbation in airway infection results in inflammatory insults that are associated with a stepwise decline in forced expiratory volume in the first second (FEV1) and quality of life. Repeated exacerbations, in turn, are associated with an increased rate of lung function decline [22]. Utilizing our current understanding of lung disease early in life and the influence of infection and inflammation, the role of proactive pulmonary surveillance and interventional strategies aims to minimize these steps of decline.

Susceptibility to infections in CF occurs mainly in the airways and not at other sites despite the presence of thickened secretions, largely as a result of the respiratory epithelium being so exposed to the external environment and, therefore, the ever-present risk of respiratory virus, bacterial and fungal microorganisms. Additionally, a variety of other factors further exacerbate this risk. The “low-volume hypothesis” paradigm revolves around the reduction of chloride and bicarbonate secreted into the airway lumen by the dysfunctional CFTR channels in the surface epithelial cells, submucosal glands, and alveolar epithelia. The epithelial sodium channel, now unopposed by reduced CTFR activity, absorbs water and sodium, further dehydrating the airway surface liquid (ASL) while increasing height of luminal mucus plugs, thereby dramatically compromising mucociliary defenses [23]. The hyper-acidification of the ASL impairs its usual antimicrobial and biofilm prevention activities [24]. Lastly, weakened primary innate hose defenses and over-exuberant adaptive immune responses for infection that cannot be eliminated give rise to an environment favorable for the establishment of chronic progressive infection, biofilm formation, and structural lung disease [25,26,27,28].

Endothelial cells are a single layer of cells lining the entire circulatory system including capillaries, and they play a central role in vascular tone, homeostatic function, and inflammation [29]. The endothelial surface layer, when activated, signals leukocyte–endothelial adhesion molecules, leading to the recruitment of immune cells. In CF, the inflammatory response is more pronounced and sustained, with elevated levels of E-selectin, IL-1β, IL-2, IL-6, and intercellular adhesion molecule 1 observed [30,31]. This dysregulated and increased inflammatory response subsequently results in the activation of polymorphonuclear neutrophils in the CF airways. Additionally, nonfunctional CTFR disrupts glycocalyx, a vasoprotective endothelial barrier lining the lumen of blood vessels [32]. Patients with CF demonstrate immoderate levels of angiogenesis, which maintains chronic inflammation [33]. Increased levels of vascular endothelial growth factor were seen in the CF lung explants [34]. Currently, the loss of CFTR function in the endothelial layer is exhibited most prominently in the lung; however, as life expectancy increases, this dysfunction may manifest through cardiovascular disease, cancer, and other vascular pathologies [35].

Overall, *Staphylococcus aureus* represents the most common bacteria colonizing the CF airway early (preceding clinical symptoms) and later leading to acute and chronic infection syndromes. *Haemophilus influenzae* (non-typeable) is also seen frequently as a cause of exacerbations in the earlier years of life, although its role in progressive airways inflammation is less clear [36]. *Pseudomonas aeruginosa* follows after the early microorganisms and is seen in greater frequency with advancement of age [15]. Early treatment of continuous anti-staphylococcal therapy may increase the risk of colonization with *Pseudomonas aeruginosa* [37]. Infections with *Pseudomonas aeruginosa* are associated with a more rapid decline in FEV1 and greater mortality [38]. Several factors exist, making it difficult to eradicate *Pseudomonas aeruginosa*. CF epithelial cells have increased adherence of piliated *Pseudomonas aeruginosa* than wildtype CFTR cells [39]. This has been postulated to be due to increased asialoganglioside-1 receptors, arising from the hyper-acidification of the trans-Golgi network in epithelial cells [40]. Additionally there are fewer clearance sites for *Pseudomonas aeruginosa* due to abnormal CFTR [41]. Persistent mucus hypersecretion in the ASL elevates the CF epithelial oxygen consumption, which generates step hypoxic gradients. *Pseudomonas aeruginosa* adapts comfortably in this oxygen-depleted environment with increased alginate expression and formation of microcolonies. As the bacterial density increases, the mucus layers become more hypoxic, further resisting host defenses and, thus, resulting in chronic airway inflammation [42].

*Pseudomonas aeruginosa* possesses its own phenotypic plasticity to drive initial infection toward a persistent one. Isolates of *Pseudomonas aeruginosa* in CF can exhibit a loss of flagella-dependent motility, increased auxotrophy, loss of O-side chains, and distance acylation of lipopolysaccharide [43]. These features give rise to a higher resistance to antibiotics and a frequently mucoid nature, differing from *Pseudomonas aeruginosa* isolates which cause acute illness in other settings. Additionally, the presence of biofilms, well-structured consortia of bacteria, embedded in a self-generating polymer matrix consisting of proteins, polysaccharides, and DNA, allow *Pseudomonas aeruginosa* to resist phagocytosis, evade the innate and adaptive immune system, and establish chronic infection [44]. The low metabolic activity and increase in doubling times of bacterial cells are responsible for the inherent resistance to antibiotic therapy, and difficulty to eradicate biofilm infections is even seen in the immunocompetent [45]. These sessile communities of bacteria ultimately stimulate an exaggerated antibody response, which leads to a persistent immune complex-mediated inflammation [46].

Although *Pseudomonas aeruginosa* and *Staphylococcus aureus* may dominate the respiratory microbiota for some patients, those with less advanced disease demonstrate a tremendous diversity of both anaerobic and aerobic organisms. These resonant pathogens of the lower airways in CF commonly include *Streptococcus, Prevotella, Rothia, Veilonella, Acitnomyces*, and *Gamella* [47]. As the CF lung disease progresses and antibiotic usage increases, diversity in the microbiome decreases. While some commensals are ostensibly benign, some may modulate virulence with other pathogens or even directly exhibit virulence [48]. The true potential and function of this microbiome and a great deal about its qualitative and quantitative heterogeneity remain to be determined.

There are a group of organisms that tend to occur later on in the course of CF lung disease. *Burkholderia cepacia* is difficult to treat and is associated with an increase mortality [49]. *Stenotrohamonas maltophilia* is seen more frequently; however, the significance of this remains unknown [50]. The prevalence of nontuberculous *Mycobacteria* (NTM) has been increasing worldwide [51]. The most common species isolated in the respiratory secretions were *Mycobacterium avium* complex, followed by *Mycobacterium abscessus*. Following repeated course of antibiotics and the development of chronic structural lung disease, fungal colonization occurs in late disease [52]. The most frequently cultured fungi are *Candida* spp., but they are rarely pathogenic. *Aspergillus* spp. are found in approximately 25% of patients with CF and can lead to a complex hypersensitivity reaction—allergic bronchopulmonary aspergillosis. *Scedosporium apiospermum* isolates are modest in CF, and their significance is unknown [53]. Distinguishing a truly pathogenic bacterium from the commensal microbiota remains a challenge. Features of an organism contributing to airway disease should be differentiated from colonization based on repeated positive cultures, temporal associations with respiratory symptoms and/or radiological manifestation of the isolate, and a critical evaluation of therapeutic response to specific antimicrobial intervention.

Viruses are a common cause of acute exacerbation and are likely to play a significant role right from birth; however, it is unclear whether severity of illness is mediated by direct viral effects or potentiation of bacterial colonization [54]. Picornavirus, influenza A, and respiratory syncytial virus are all major pathogens reported in CF exacerbations, especially during colder seasons [55]. Respiratory viruses leading to pulmonary exacerbations among people with CF show a clear association with decline in lung function and risk of death [56]. In recent times, the COVID-19 pandemic, including the subsequent rollout of an effective vaccination program, complemented by the impact of social distancing and personal hygiene measures on the spread of all respiratory viruses at different stages of the pandemic and the dramatic positive impact of CFTR modulator therapies for those now on these medications, have resulted in highly variable healthcare utilization and morbidity in CF patients. Overall, CF patients post lung transplantation have arguably been proven to be the most at-risk patients in all phases of the COVID-19 pandemic so far [57]. Although the pandemic also impacted clinical research by limiting active trials and specialized CF care programs, this was thankfully kept to a minimum in most countries where these activities were well established [58].

Fascinatingly, at the other end of the spectrum, CF carriers were found to be disproportionately represented in cohorts of severe COVID-19 disease requiring mechanical ventilation when compared to noncarriers [59]. Only time will tell whether this also relates to an inherent bias toward a proinflammatory state in these patients, which is counterproductive in acute viral infection where there is minimal or no pre-existing immunity. One cannot help but notice, however, that this speculation sits well with the current evidence indicating that CF carriers are known to have an increased predisposition for asthma [60], and that airway infection in CF is accompanied by exaggerated inflammation with an excess of proinflammatory cytokines and alterations in innate and adaptive immune responses. Not only has it been shown that CF airway macrophages are hyperinflammatory, a phenotype that perhaps is independent of defective CFTR chloride channel function [61], but it is also highly possible that an overly exuberant innate immune/inflammatory response and influx of proteases may maladaptively drive bronchiectatic changes by both leading to “unchecked” tissue damage and diminishing the restorative capacity of the lung.

Abnormal CFTR protein results in both impaired chloride transport across cell membranes and direct/indirect uncoupling of key cell membrane-associated signaling processes that are associated with CFTR-dependent tight phospholipid raft formation, as schematically represented in Figure 2 [62].

Additionally, the disorganized intracellular dynamics of CFTR processing, which includes abnormalities in protein synthesis, folding, and trafficking, not only leads to reduced functional CFTR protein reaching the cell membrane but also dramatically increases the metabolic burden of affected cells. Significantly misfolded or dysfunctional CFTR is redirected to the proteasome for degradation and recycling [63]. Ubiquitination of abnormal proteins targets them for the proteasome where degradation occurs [64]. Abnormal CFTR proteins (large complex proteins) requiring degradation via the proteasome necessitate an increase in the metabolic requirements of the cell. It is hypothesized that CFTR both “burdens” and “blocks” the proteasome. This impacts multiple processes, including antigen processing, cell cycle and division, apoptotic pathways, the modulation of cell surface regulator/channels and secretory pathways, and the metabolic requirements of the cell. Notably, the effect of CFTR dysfunction may differ in nonepithelial cells. Neutrophils in CF patients (F508del) were observed to have a prolonged survival and reduced degranulation, affecting the ability to kill bacteria effectively [65].

There are no doubt pathogens that are unable to be specifically identified using standard microbiological techniques and perhaps even despite the broad range of research technologies currently available. There is also now a much richer awareness and recognition of the interplay among various microbes in either the chronic biofilm setting or acute interaction settings. Additionally, the role of innocent commensals, synergins, and true pathogens may oscillate on the basis of host factors, such as immune deficiencies, structural lung disease, environmental triggers, bacterial signaling, and/or exogenous viruses.

Lastly, the world environment poses a further threat to the perpetuation of any infection-amplified lung inflammation. Outdoor air pollution, including biomass burning and traffic, impacts lung function and exacerbation frequency via oxidative stress [66]. Regions with a warmer climate are associated with lower FEV1 in CF, likely mediated by the higher prevalence of specific microbes [67]. Indoor air pollutants have not been studied in the CF population, but fireplaces and gas furnaces have been associated with morbidity in children with asthma [68]. Those exposed to secondhand tobacco smoke have been shown to impair CTFR in non-CF patients both in vitro and in vivo [69]. Higher rates of methicillin-resistant *Staphylococcus aureus* are seen in infants with CF who have parental smoke exposure [70]. Although CF is a genetic condition, factors such as the socioeconomic status, household characteristics, and geographical location in the world environment can create vulnerability and variability in disease progression and health outcomes.

## 3. The Evolutionary Past (Last 5000–50 Years)

CF affects more than 70,000 individuals worldwide [12,71]. The estimated incidence in the United States white population ranges from one in 1900 to one in 3700 [72]. CF is less frequent in Asians, African Americans, and Hispanics [73,74]. It results from mutations in a single gene on the long arm of chromosome 7 and consists of 27 exons that encode the 1480-amino-acid CFTR protein [23]. CFTR characteristically comprises two nucleotide-binding domains, two membrane-spanning domains, and a unique regulatory R domain with multiple phosphorylation sites, and it is a member of the ATP-binding transporter family of membrane proteins [75,76]. One clear consequence of reduced CFTR function is the altered transportation of chloride and bicarbonate at the apical membrane of the epithelial cells.

More than 2000 mutations have been reported in CFTR, with over 400 known to cause CF [77]. The first identified and most common mutation is the deletion of three base pairs at position 508 of CFTR that codes for phenylalanine (Phe508: F508del). This is a class II mutation. Homozygous F508del and heterozygous F508del are the most commonly seen variations at 47% and 43%, respectively. The remaining mutations are highly heterogenous, with fewer than 20 mutations occurring at a frequency higher than 0.1% [77]. After F508del (72%), the most common individual allele variants captured in the Australian registry include G551D (4.2%), R117H (1.8%), G532X (1.6%), and 1717_1G_>A (1.0%). The remaining variants comprise less than 1.0% each [12]. Currently, CF mutations are classified into six different functional classes, each based on the amount of cell surface protein, channel stability, and function [23] (Table 1). Class I mutations cause defective production of the protein, resulting in a total loss of function for CTFRs. Class II mutations cause defective protein, leading to CFTR in an incorrect location in the cell or a reduction in glycoproteins/gangliosides on the cell surface. Class III and IV mutations cause defective regulation of channel opening and ion conduction through CFTR, respectively. They are typically associated with a less severe phenotype and normal pancreatic function [78]. Class V mutations are insufficient protein variants which result in a reduced amount of CFTR at the membrane surface. Class VI mutations result in reduction in membrane retention because of CFTR instability.

Despite the limited life expectancy, the carrier rate of the CFTR gene remains ever present. An estimated one in 25 Australians carries the mutation [12]. Although the true reason for this high carrier frequency remains unknown, various factors and explanations are likely to be causally interacting. Most proximally, a founder effect accounting for the amplification of allele frequency in the emigrant population from Western Europe in particular, as an extension of similar observations involving other specific communities such as the Amish, has been suggested [80]. Further in the past, selective enrichment of specific gene mutations in Western Europe that have occurred at random throughout history may be at play [81].

It has been previously postulated that the most likely selection pressures for this heterozygote advantage are life-threatening infectious diseases [82]. Indeed, reduced CFTR function in epithelial cells may confer protection against cholera and typhoid fever by limiting the fluid loss during these illnesses [83,84] The evidence supporting this is circumstantial at best and epidemiologically relatively weak. Secretory toxins of enterotoxigenic *Escherichia coli* and *Vibrio cholerae* act via CFTR to transport chloride into the intestinal lumen, causing osmotic diarrhea. [85]. Murine models have demonstrated that the absence of CFTR results in no fluid secreted in response to a toxin challenge [86], and a thiazolidinone-type CFTR inhibitor has indeed been developed to reduce toxin induced fluid and ion loss in secretory diarrheas [87]. On the other hand, a dramatically lower incidence of *Mycobacterium tuberculosis* (TB) has been observed in heterozygotes of CF [88]. Detailed epidemiological analysis mapping the geographic prevalence of CF carrier rate patterns with the history of known infectious disease epidemics suggests that perhaps the selective pressure exerted by TB may be a particularly strong factor for the high rates of CF in the industrialized countries of western Europe in particular, as well the countries that these populations emigrated to over the last 300 years [81,89]. One potential reason for this may stem from the diminished activity of arylsulfatase B seen in patients with CF. Arylsulfatase B, also known as N-acetyl galactosamine 4-sulfatase, is a sulfatase enzyme that is responsible for the degradation and desulfating of sulfoglycosaminoglycans. The survival and pathogenicity of TB are related to the sulfation of cell-wall glycolipids; hence, diminished lysosomal arylsulfatase activity in CF may be a vital defence mechanism, by limiting the supply of available sulfate for the synthesis of the cell wall [90]. This also results in the CF microbiome being rich in sulfated polysaccharides, favoring the colonization of bacteria such as *Pseudomonas aeruginosa* and *Burkholderia cepacia* which uses them as substrates [12].

All of the above raises the possibility that, although the molecular mechanisms of severe CFTR dysfunction may significantly predispose the host to progressive cycles of chronic infection-associated bronchiectatic lung disease (classic CF disease), much milder levels of CFTR dysfunction (not usually associated with abnormal sweat chloride losses) may have proinflammatory innate immune benefits that were particularly relevant in the pre-antibiotic era. Even more recently, CF carriers appear to have a higher risk of developing severe coronavirus disease 19 (COVID-19) [59].

## 4. The Long Now (Last 50 Years)

The survival advances in CF that we have seen in the recent past (i.e., the long now) can largely be attributed to the comprehensive CF care systems that have gradually evolved over the last 50 years to manage the phenotypic consequences of the CF genotype and their overall orchestration through focused, dedicated, and expert multidisciplinary teams [91].

The basic tenets of treating CF-associated airway infection include targeting recently identified isolates often with combination antimicrobial therapy, adopting an integrating review of antimicrobial susceptibility profiles and previous clinical therapeutic response patterns, and aiming for clearly defined endpoint-driven long-term antibiotic use strategies—all designed to maximize efficacy whilst minimizing antibiotic side-effects and resistance development. With this in mind, early “eradicating” antipseudomonal strategies for first isolates of *Pseudomonas* can be very beneficial in delaying the establishment of chronic infection; however, if unsuccessful, prolonged use may exacerbate the evolution of new, genetically different *Pseudomonas aeruginosa* isolates with more resistant antibiotic profiles [92]. Indeed “early” *Pseudomonas* eradication failure rates have been reported as high as 75% at 12 months post therapy [93]. Nevertheless, once chronically colonized, maintenance antibiotic therapy in selected patients with frequent exacerbations in airway sepsis has been associated with lung function stabilization and reducing morbidity [94]. The addition of nebulized aminoglycosides has been particularly beneficial in this setting, resulting in significant reductions in pulmonary exacerbation rates and greater improvements in FEV1 stabilization [95]. Long-term azithromycin is now also used for its anti-inflammatory and anti-infective properties in selected CF patients [96]. This macrolide is able to accumulate within neutrophils, impairing its inflammatory cytokine and oxidant production. A reduced rate of pulmonary exacerbations and increased weight were seen following treatment for 6 months [97]. Nebulized therapy, including dornase alpha hypertonic saline, as well as various airway clearance techniques, has also been shown to be beneficial both alone and in combination with antibiotic treatment strategies, in removing and reducing infected mucopuruent secretions [98,99].

When acid-resistant pancreatic enzymes became available in the 1980s, a rise in nutritional status led to better nutritional and overall long-term outcomes [100]. As CF lung disease increases, an increased calorie requirement due to increased airway sepsis, inflammation, and work of breathing also occurs. In addition to a high-protein unrestricted caloric diet being necessary to maintain nutritional needs in a highly catabolic CF patient, maintaining adequate hydration and salt balance is also imperative to maintain adequate cellular, tissue, and organ homeostasis, thereby further helping to prevent lung function and maintain health stability. This proactive approach targeting the early identification of nutritional “risks” and pre-emptive intervention strategies to mitigate against have gradually become the standard of care in CF centers of excellence both in the acute setting when CF patients are hospitalized for pulmonary exacerbations and in the long-term, ambulatory care setting.

Coordinated and aggressive multidisciplinary team management for all aspects of CF-associated complications has improved CF care by alleviating symptoms and improving morbidity, quality of life, and long-term prognosis, despite none of these therapies specifically addressing the core problem—the need to restore CFTR function.

## 5. The Present Moment (Last 5 Years)

Most recently, tremendous advances in CF therapy have led to the development of small molecules (CFTR modulators) targeting the CFTR protein, thereby paving the way to hope for a cure for CF [101]. In general, CFTR modulators can be classified as correctors, potentiators, stabilizers, or amplifiers. Correctors aim to stabilize the misfolded protein the cytosol to prevent degradation. Potentiators bind to the nucleotide-binding domain of the CFTR channel to facilitate its opening. Stabilizers rescue protein stability on the membrane and promote CFTR maturation. Amplifiers are small molecules which increase the amount of CFTR production. The first successful modulator—ivacaftor (Kalydeco), a potentiator, was originally developed for the individuals with at least a copy of a class III mutation (Gly551 Asp-CFTR) [102]. Landmark trials showed reduced sweat chloride, increase in FEV1, weight gain, reduced frequency of pulmonary exacerbations, and quality-of-life improvements in adult patients. These findings were replicated in those aged 12 and above, with effects observed for up to 144 weeks [103,104]. Subsequent progress saw a combination of ivacaftor and lumacaftor (Orkambi), an oral corrector agent which in vitro corrects misprocessed protein leading to increased cell membrane localized protein, targeting both CFTR trafficking and channel activation in patients with two copies of F508del. While similar outcomes were observed as compared with ivacaftor monotherapy, the absolute FEV1 increase was modest [105].

Furthermore, respiratory-associated side-effects largely driven by an increased propensity to airway reactivity was problematic, and real-world use was associated with high discontinuation rates [106]. These shortcomings drove further development and led to a much-improved corrector in tezacaftor, which had more advantageous pharmacological properties and a better side-effect profile [18]. The combination of ivacaftor and tezacaftor (Symdeko/Symkevi) was clinically effective and well tolerated in both F508Del homozygotes and F508Del heterozygotes with a concomitant residual function mutation [107].

While single and dual therapies were not suitable for a substantial proportion of CF mutations (only G551D and F508Del homozygotes qualified), it did provide a successful roadmap for subsequent generation correctors used in combination with tezacaftor/ivacaftor. Phase 3 studies using elexacaftor (CFTR corrector) together with tezacaftor/ivacaftor (Trikafta/Kaftrio) elicited a 10% increase in FEV1 predicted for F508Del homozygotes [108]. The F508Del minimal function cohort (no CFTR protein produced or a CFTR protein that does not respond to ezacaftor, ivacaftor, or tezacaftor/ivacaftor in vitro) saw a greater increase in FEV1 predicted at 14.5%, with a similar increase in quality-of-life measures, sweat chloride, and pulmonary exacerbation reduction [8].

CTFR modulators also have the potential to exert their effects beyond improvements in sweat chloride channel function and the pulmonary system. Surrogate markers of pancreatic function, fecal elastase, and nutritional status were significantly improved following 24 weeks of ivacaftor therapy [109]. The frequency of pancreatitis was also reduced and may be related to increases in duodenal pH [110]. The prevalence and relative risk of CF related diabetes were observed to be lower in large registry data, although its effect on the insulin-secreting β-cell is unclear [111]. CFTR is expressed solely by cholangiocytes lining the bile duct epithelium, and ductal obstruction leads to the complex sequalae of CF-associated liver disease [112]. Large observational studies indicated a lower number of hepatobiliary complications when compared to those not on modulator therapy; however, a treatment effect was not directly established [111]. There is also mounting evidence of its effect on chronic rhinosinusitis, neurocognition, and fertility [113].

At an immuno-inflammation and host defense level, CFTR modulators have created a new chapter in correcting dysregulated immunity. Ex vivo studies have found increased oxidative burst and neutrophil activity, with subsequently improved phagocytosis and enhanced killing of *Pseudomonas aeruginosa* in patients on ivacaftor [114,115]. An anti-inflammatory effect has also been demonstrated. Reductions in endoplasmic reticulum stress, neutrophil–epithelial cell binding, and inflammatory mediators (CXCL7, CXCL8, and sTNFR1) were reported [116]. Other cell lineages have also shown decreased inflammatory markers from patients on ivacaftor [117]. Much is uncertain as to what degree these effects are a direct result of CFTR modulators on specific innate and adaptive immune cell function or due to indirect improvements in mucociliary clearance, nutritional status, and indeed general metabolic functions that remain poorly resolved.

Access and price to CFTR modulators differ largely on the basis of geographic location, government subsidies, and funding. Licensing of the same medication between the United States and Australia had almost a 18 month gap [118]. To this day, some nations have no access to CFTR modulators [119]. Addressing the barriers to this inequality would be a challenge that requires campaigning from policymakers, professional associations, and healthcare leaders.

## 6. The Near Future (Next 5 Years)

Better targeted CF treatment interventions, along with improved service delivery systems, have given us a unique opportunity to reimagine CF care such that it can now cater to greatly increased expectations of wellbeing and psychosocial functioning, as well as an increasing prevalence of many previously less common age-related complications. As the rates and severity of CF-associated pulmonary disease improve, the importance of gastrointestinal, metabolic, endocrine, hepatobiliary, and neuropsychological aspects of CF are likely to move to the forefront [120]. Additionally, as survival improves and the number of patients on CFTR modulators rise, the long-term unknown side-effects of these therapies are likely to become more evident. Looking toward the future, we would anticipate surveillance and CF registry programs to more broadly and more precisely elucidate both improved outcomes and the potential for adverse events.

Given all the recent changes in CF care strategies, the likely outcome benefits, and the potential for “new” complications, different models of care will be required to meet the changing needs of our patients. In particular, the incorporation of telehealth access and monitoring strategies into more standard ambulatory care models to cover a “larger, healthier” CF population as they get older (with a particular multidisciplinary team care focus) will need to be integrated with maintaining the specialist expertise and resource to manage both this CF care setting and a relatively smaller group of more complicated inpatients where the focus of the multidisciplinary team care is likely to be very different.

Lastly, clinical trial pipelines are continuing to actively explore many other approaches targeting both CF genotypes abnormalities associated with no CFTR protein production and non-CFTR therapeutic pathways, which will be increasingly needed as Class I mutations (~22%) and rare CF variants are not expected to benefit from current CFTR modulators [79].

## 7. Future Horizons

Earlier commencement of CFTR modulators is likely to yield benefits in morbidity and mortality, which are closely linked to reduced acute lung disease exacerbation rates and much fewer episodes of lung function decline throughout a CF patient’s life journey [121]. The future may see further studies to validate the use of elexacaftor/tezacaftor/ivacaftor for F508del homozygotes or heterozygotes from as early as 2 years old and maybe even from birth. Little is known about the relationship between maternally ingested CFTR modulators and the fetus, although there is evidence that newborn CF babies exposed to CFTR modulators in utero have a normal CF screening test and sweat chloride [122,123]. The Therapeutic Goods Administration in Australia current classifies all CFTR modulator therapy as category B3, which advises caution when continuing treatment throughout pregnancy [124]. In the case of a CF mother and CF baby or non-CF mother and CF baby, it is unclear if CFTR modulators before birth would confer even greater benefits by correcting mutations in utero for at-risk organs before pulmonary and extrapulmonary complications arise. While these possible benefits will clearly need to be balanced against the risk of potential unknown on- and off-target side-effects including a potential impact on fetal brain development, the question of running clinical trials on pregnancy-associated CF subgroups will always be ethically very challenging. For those with CF not eligible for CFTR modulator treatment, hope may lie in gene therapy—a developing novel treatment to integrate a new and correct copy of CFTR gene to deoxyribonucleic acid [125]. Predominantly used in the oncological field at this moment, safety and delivery have improved, and gene therapy may yet again come into the spotlight.

By correcting CFTR dysfunction and improving both mucociliary clearance and “tightening” inflammatory signaling across cellular phospholipid membranes, CFTR modulator therapy may reduce the inflammatory burden of typical CF pathogens, creating a new airway microbiome [126]. Ivacaftor therapy reduced the relative abundance of *Pseudomonas aeruginosa* and saw the reciprocal rise of other commensal organisms such as *Streptoccus, Prevotella, Veillonella*, and other taxa. Other studies observed a rebound in *Pseudomonas* in later samples or no changes at all [127]. There is little known about the changes in biofilm composition of *Pseudomonas* in the presence of modulators. Furthermore, changes in antibiotic exposure while on modulator therapy to treat infections were associated with changes in sputum microbiota composition [128]. The effect of CFTR modulator therapy on specific fungal and other rarer microbes is still unknown, and this is also true regarding the long-term impact of these therapies on specific virus-associated respiratory disease, although the initial signals are largely positive. Most published studies were performed when ivacaftor was the only CFTR treatment available; hence, it will be interesting to assess the differential impact—if any—of newer CFTR modulators. Additionally, examining the correlation between microbiome changes and inflammatory alterations using later-generation CFTR modulators such as elexacaftor/tezacaftor/ivacaftor in various age groups will likely provide both greater insight into underlying pathobiological interactions and, potentially, further opportunities for therapeutic intervention. Moreover, organs-on-a-chip, modern microengineered systems that mimic the physiological, functional, and physiochemical characteristics of human tissue, provide a platform to study inflammation, drug efficacy, pathogenesis, and disease model development in CF [129]. The uptake of this technology is in the early stages as the chip model undergoes further refinement in cell adhesion and mixing, as well as moves toward a 3D cell culture. However, the wealth of opportunity it can bring forth is undeniable and will be key to understanding the interaction of inflammation and infection in the years to come.

Finally, will it be possible to both provide an individualized approach to CF care that will be as value-adding as possible (integrating all potential therapies according to effectiveness whilst minimizing the overall burden of care) and not fall into the trap of overgeneralizing treatment by focusing only on a small number of modulators? Only time will tell.

## Figures and Tables

**Figure 1 ijms-24-04052-f001:**
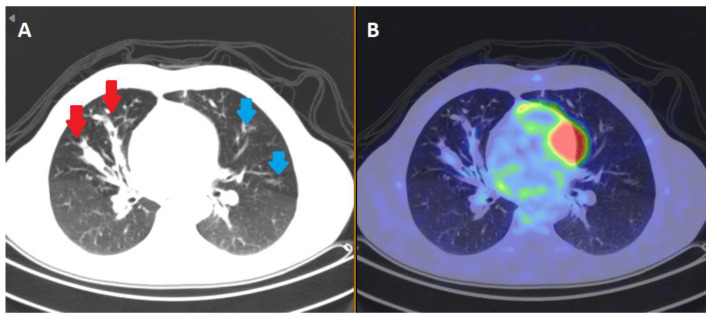
Sequalae of inflammation in CF. Various areas of inflammatory signal in the lung fields of a 40 year old male CF patient with relatively stable pseudomonal chest sepsis demonstrated on positive emission tomography (PET) body scan. Distal nodule opacities (blue arrows) and gas trapping distal to this are appreciated on the computerized tomography (CT) slice in box (**A**). The red arrows reflect bronchiectatic changes in the middle airways. In box (**B**), a positron emission tomography (PET) slice shows areas of F-fluorodeoxyglucose (FDG) tracer avidity in the corresponding areas of bronchiectasis, reflecting the infective/inflammatory nature of the lungs. Source: Alfred Hospital, Melbourne, VIC, Australia.

**Figure 2 ijms-24-04052-f002:**
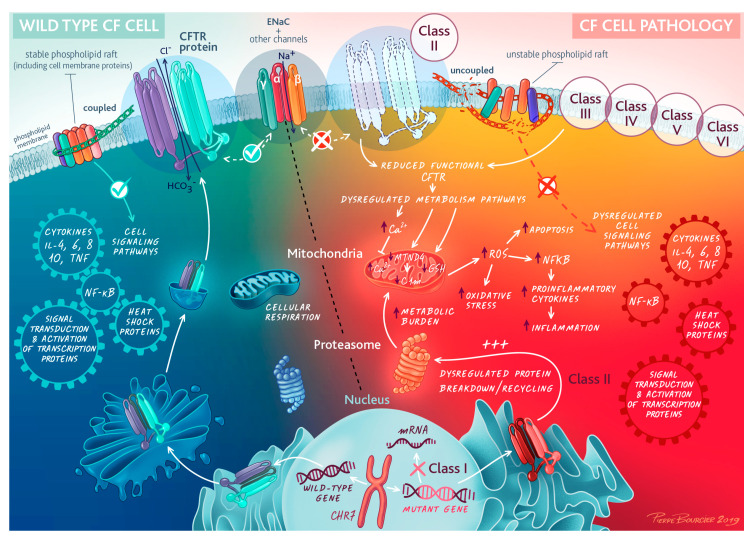
Cell biology of cystic fibrosis transmembrane regulator (CFTR). Abnormal CFTR and signaling pathways at the epithelial cell membrane. Mutation classes are discussed in depth in Section 3. Reprinted/adapted with permission from © ERS 2023: European Respiratory Journal May 2020, 55 (5) 1902443; DOI: 10.1183/13993003.02443-2019. Copyright and Licensing are available via the following link: https://www.mdpi.com/ethics#10.

**Table 1 ijms-24-04052-t001:** Cystic fibrosis transmembrane receptor (CFTR) mutation classes, functional impairment, frequency, and CFTR modulator therapies. ^α^ Occurrence of at least one mutation in that class based on United States registry data [79].

Traditional Class	CFTR Impairment	Mutations Example	Frequency of Occurrence ^α^	CFTR Modulator Treatment
I	No mRNA/functional protein	G542X	22%	None
II	No protein trafficking	F508Del	88%	Lumacaftor–ivacaftor, tezacaftor–ivacaftor, or elexacaftor–tezacaftor–ivacaftor
III	Abnormal gating of channels	G551D	6%	Ivacaftor
IV	Reduced channel conductance	R117H	6%	Ivacaftor in select mutations
V	Decreased protein synthesis	A455E	5%	Tezacaftor–ivacaftor in select mutations
VI	Reduced protein stability	c.120del23	5%	Tezacaftor–ivacaftor in select mutations

## Data Availability

No new data were created or analyzed in this study. Data sharing is not applicable to this article.

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
