# Peer review of "Respiratory Infection and Inflammation in Cystic Fibrosis: A Dynamic Interplay among the Host, Microbes, and Environment for the Ages"

_ijms, 2023, doi:10.3390/ijms24044052_

Round 1
Reviewer 1 Report
In this review, Yu and Kotsimbos provide a complete overview of the CF disease, focusing mainly in inflammation, bacterial and viral infection in addition to the problems due to the external factors as pollution. The authors also discuss gastrointestinal, metabolic, endocrine, hepatobiliary, and neuropsychological aspects and many others, in addition to the respiratory disease, which become more emergent in the recent year, with the increased life expectancy.
The authors discuss the advent and evolution of the CFTR modulators, with benefits, present side effects and complications that can arise in the future. They also provide a detailed historical summary of the CF disease, from its discovery to today, giving insights and perspectives for the near future.
The review is interesting and provide a good overview of CF.
However, there are some points that could be included and make this manuscript more complete.
1) The authors discuss the inflammation in CF and abnormal neutrophilic infiltration in the airways, however, they don’t discuss at all the problem of the endothelial dysfunction in CF Endothelial cells, which play a key role in inflammation and PMN migration, have been isolated from CF patients’ lungs and studied, showing a pro inflammatory profile and several additional dysfunctions of CF endothelium.
2) As concerns the inflammation, the authors do not discuss the importance of the 3D ex-vivo structures as the microfluidic organ-on-a-chip for studying neutrophil migration and inflammation in CF. Perhaps it might be useful to add this in the Future Horizons.
Language:
A general review of the English language is recommended
General:
Staphylococcus aureus, Pseudomonas aeruginosa and the other names of bacteria should be in italic
Author Response
1) The authors discuss the inflammation in CF and abnormal neutrophilic infiltration in the airways, however, they don’t discuss at all the problem of the endothelial dysfunction in CF Endothelial cells, which play a key role in inflammation and PMN migration, have been isolated from CF patients’ lungs and studied, showing a pro inflammatory profile and several additional dysfunctions of CF endothelium.
Response: Thank you for your comments. We have included a paragraph (lines 110-120) to emphasize the role of endothelial dysfunction in CF lung, and other vascular pathologies as the life expectancy of CF increases.
2) As concerns the inflammation, the authors do not discuss the importance of the 3D ex-vivo structures as the microfluidic organ-on-a-chip for studying neutrophil migration and inflammation in CF. Perhaps it might be useful to add this in the Future Horizons.
Response: We appreciate your time and consideration of our review paper. Thank you for highlighting this gap in our paper. We have added a paragraph (lines 456-461 ) in the Future Horizons section as suggested to ensure the reader is aware of this emerging technology.
Names of bacteria all italicized.
Reviewer 2 Report
The present review gives a historic overview of cystic fibrosis, then outlines the game-changing effect on morbidity CFTR modulators have had in recent years as well as gives a perspective for the future.
The review is well written and easy to follow and generally well structured.
I have a several minor concerns related to structure, wording and references and would like to suggest some additional information and Figures that in my opinion the review would benefit from.
Page 1
l.11: Cystic Fibrosis not cystic fibrosis
l.16: 'are now revolutionized' not 'is..'
l.17: Are CFTR modulators considered gene modulators as they affect the protein
Page 2
l. 44: CF gene - Please specify that the CFTR gene is meant here as this is the first mention of what CF is actually caused by
l. 51: Please check Refs #5-7, which don't seem to encompass interventions to improve CFTR function
l. 53-63: This paragraph could benefit from briefly mentioning recurrent infections, lung inflammation and lung damage, especially given that the paragraph concludes by saying that pulmonary disease is the prime cause of effect
l.54: 'This function is lost' .. It can also be diminished as pointed out later in the review when discussing the different classes of CFTR mutations.
l. 66: '..95% oF CF patients eligble for these agents' - Out of how many total CF patients?
l.67-68: This would be easier to understand if it had first been mentioned that different classes of mutations exist, also see comment above.
l. 72: Again, there is mention of heterogenous CF cohorts without having mentioned the different mutation classes. In addition to mentioning that these exist earlier, the authors could refer to the later section of the review. But as a reader, I was looking for this information not knowing that this was going to be dealt with later in the review.
Page 3
l. 8: Please double-check reference 17, which seems to be on smoke exposure.
The following reference should be considered:
N Engl J Med 2013; 368:1963-1970
DOI: 10.1056/NEJMoa1301725
l. 89: Please double-check reference 18
l. 107: Please use singular 'bacterium' or alternatively use 'are'
l. 110: Please double-check reference 27. This refernece describes increased colonization with PsA following anti-staphylococcal treatment.
Page 4
l. 136: I would prefer the use of 'pathogen' instead of 'bug'
Figure: The relationship between the text and the Figure is unclear. How does the Figure relate to microbiome heterogeneity? Also the Figure legend is not sufficiently explained. What is FDG-avidity? What is the source for this image? What are the different subpanels? Where in the image is the reader to see things mentioned in the legend, such as distal nodule opacities and gas trapping etc..
l. 148: Consider rewording the sentence: 'focal moderately increased FDG uptake...'
Page 5:
l. 156: Please include a reference for S. maltophilia.
l. 190: Like with Figure 1, the relationship between the text here and Figure 2 is not that clear. A lot of details are in the legend that are not discussed in the text and are missing references. Such as phospholipid raft formation and proteasomal degradation and its associated metabolic demand. These should be properly cited and placed in the main text.
Page 6:
Figure 2: Different classes of CFTR are shown here, but aren't yet mentioned in the text. Consider making a note in the legend to the later section of the text. Please highlight tat the pathways here are related to epithelial cell membranes. This might be different for CFTR on leukocytes. Please briefly discuss the effect of CFTR mutations on leukocyte function including neutrophils elsewhere in the text and cite recent reviews for further reading. Should the label read 'Wild type CF cell'? Or just WT Cell?
l. 193: replace 'receptor' with 'regulator'
Page 7:
l. 223: Please include a reference after 'in vivo'. Does it matter whether smoke impairs CFTR function if it is already impaired in CF due to mutation? Consider rewording if this is not what was meant here.
l. 229: It seems odd to lump Australian and global statistics together, then list other individual countries afterwards.
l.238-251 Given the Phe508del is the most common mutation, does it pay to mention which class it belongs to?
Page 8:
l. 268: Consider moving this paragraph to the start of the next paragraph and also merging the paragraph starting at l. 276, so that everything on TB is in one paragraph.
l. 276: Do the authors mean carriers? I guess this would be less well studied than CF patients. Please clarify.
l. 277: enzyme that 'is', not 'are'
l. 282: This sentence seems to not support the argument for why carriers have persisted in the population?
l. 302: Remove full stop after 'indeed'.
Page 9:
l. 309: Include the recent clinical trial: DOI: 10.1016/S2213-2600(22)00165-5
Page 10:
l. 348: Explain why the single and dual therapies weren't suitable for a substantial proportion of CF mutations.
l. 350: What type of modulator is elexacaftor?
l. 351: What do the authors mean by minimal function cohort?
l. 371: Can the authors briefly comment on limitations around price, access, funding and approval in different countries?
Page 11:
l. 392: How many of total CF patients are affected by Class I?
l. 401: Again, I am confused why CFTR modulator therapy is classified as gene therapy.
l. 394: Also in this section, can the authors speculate on the future advances in CF care using gene therapy, i.e. introducing a copy of the functional gene?
l. 425: Add a hyphen between 'value' and 'adding'.
Other suggestions for improvement:
A lot of what is discussed in the review and the benefit of various modulator therapies relies on understanding the different classes of CFTR mutations. Consider having a table with the different classes, their functional impairment category, examples of mutations, frequency of occurance and effective treatment regime.
Another Figure could show a scheme of the main points that characterized each chapter of the CF history/treatment/care given that this is the focus of the review as opposed to numerous other reviews that show similar Figures as that in Figure 2.
Author Response
Response: Thank you for reviewing our manuscript. We have edited the paper based on your recommendations.
Page 1
l.11: Cystic Fibrosis not cystic fibrosis
Response: This has been corrected
l.16: 'are now revolutionized' not 'is..'
Response: This has been corrected
l.17: Are CFTR modulators considered gene modulators as they affect the protein
Response: 3, gene has been removed to avoid confusion.
Page 2
- 44: CF gene - Please specify that the CFTR gene is meant here as this is the first mention of what CF is actually caused by
Response: Sentence changed to specify location of gene on chromosome 7 and mutation F508Del identified.
- 51: Please check Refs #5-7, which don't seem to encompass interventions to improve CFTR function
Response: Thank you for identifying this. We have referenced 2 further articles.
Middleton, P.G., et al., Elexacaftor–Tezacaftor–Ivacaftor for Cystic Fibrosis with a Single Phe508del Allele. New England Journal of Medicine, 2019. 381(19): p. 1809-1819.
Barry, P.J., et al., Triple Therapy for Cystic Fibrosis Phe508del–Gating and –Residual Function Genotypes. New England Journal of Medicine, 2021. 385(9): p. 815-825.
- 53-63: This paragraph could benefit from briefly mentioning recurrent infections, lung inflammation and lung damage, especially given that the paragraph concludes by saying that pulmonary disease is the prime cause of effect
Response: We have added 2 sentences mentioning pulmonary exacerbations, lung inflammation and infection. The paragraph flows better now.
We did not go in-depth into this as it was further discussed in ‘The Host and the Environment: Inflammation and Infection”.
l.54: 'This function is lost' .. It can also be diminished as pointed out later in the review when discussing the different classes of CFTR mutations.
Response: We have corrected this ‘or diminished’ after lost.
- 66: '..95% oF CF patients eligble for these agents' - Out of how many total CF patients?
Response: We have restructured this to 90-95% of the Australian CF populations of over 3600 patients.
l.67-68: This would be easier to understand if it had first been mentioned that different classes of mutations exist, also see comment above.
Response: Sorry for the confusion. We have added ‘for various class mutations’ in the first sentence to ensure the reader is aware that different class mutations exist.
- 72: Again, there is mention of heterogenous CF cohorts without having mentioned the different mutation classes. In addition to mentioning that these exist earlier, the authors could refer to the later section of the review. But as a reader, I was looking for this information not knowing that this was going to be dealt with later in the review.
Response: We apologize for this, For the first paragraph (A Brief History), we wanted to give an overview without going into the fine details but understand how it is frustrating for the reader. We have added the ‘Mutation classes discussed in depth in The Evolutionary Past section’ at the end of that paragraph to ensure the reader is aware this will be covered later on.
Page 3
- 8: Please double-check reference 17, which seems to be on smoke exposure.
The following reference should be considered:
N Engl J Med 2013[1]; 368:1963-1970
DOI: 10.1056/NEJMoa1301725
Response: Apologies, the reference has been updated to suggest article.
- 89: Please double-check reference 18
Response: This reference has been updated to
Berger, Melvin. “Inflammatory mediators in cystic fibrosis lung disease.” Allergy and asthma proceedings vol. 23,1 (2002): 19-25.
- 107: Please use singular 'bacterium' or alternatively use 'are'
Response: Changed to ‘are’.
- 110: Please double-check reference 27. This reference describes increased colonization with PsA following anti-staphylococcal treatment.
Response: Thank you for pointing this out. We have deleted that sentence and added Early treatment of continuous anti-staphylococcal therapy may increase the risk of colonization with Pseudomonas aeruginosa with the same reference in the subsequent sentence line
Page 4
- 136: I would prefer the use of 'pathogen' instead of 'bug'
Response: Changed to pathogens
Figure: The relationship between the text and the Figure is unclear. How does the Figure relate to microbiome heterogeneity? Also the Figure legend is not sufficiently explained. What is FDG-avidity? What is the source for this image? What are the different subpanels? Where in the image is the reader to see things mentioned in the legend, such as distal nodule opacities and gas trapping etc..
Response: Thank you for your comments. We have cut down the figure to 2 images. The CT scan and corresponding PET slice. The paragraph has been rewritten to articular the images, include F-fluorodeoxyglucose (FDG) tracer full name and arrows. We hope this makes the image and legend clearer.
- 148: Consider rewording the sentence: 'focal moderately increased FDG uptake...'
Response: This has been reworded in the restructure of the figure – in the comment above.
Page 5:
- 156: Please include a reference for S. maltophilia.
Response: Upon further review of S. maltophilia in CF, it appears the significance is unknown. Recent studies show it does not confer a worse prognosis. Sentence has been restructured and reference added.
Goss, C.H., et al., Detecting Stenotrophomonas maltophilia Does Not Reduce Survival of Patients with Cystic Fibrosis. American Journal of Respiratory and Critical Care Medicine, 2002. 166(3): p. 356-361.
- 190: Like with Figure 1, the relationship between the text here and Figure 2 is not that clear. A lot of details are in the legend that are not discussed in the text and are missing references. Such as phospholipid raft formation and proteasomal degradation and its associated metabolic demand. These should be properly cited and placed in the main text.
Response: The bulk of text in figure has been moved to the main text and cited accordingly.
Page 6:
Figure 2: Different classes of CFTR are shown here, but aren't yet mentioned in the text. Consider making a note in the legend to the later section of the text. Please highlight that the pathways here are related to epithelial cell membranes. This might be different for CFTR on leukocytes. Please briefly discuss the effect of CFTR mutations on leukocyte function including neutrophils elsewhere in the text and cite re[2]cent reviews for further reading. Should the label read 'Wild type CF cell'? Or just WT Cell?
Response: Thank your for these comments. Note added to mention about mutation classes discussion later in the text. Added ‘signalling pathways at the epithelial cell membrane’ to ensure location is clear. CFTR effect on leukocyte function added – article referenced
Averna, M.; Melotti, P.; Sorio, C. Revisiting the Role of Leukocytes in Cystic Fibrosis. Cells 2021, 10, 3380. https://doi.org/10.3390/cells10123380
Label reads ‘Wild type CF cell’ as intended.
- 193: replace 'receptor' with 'regulator'
Response: This has been replaced.
Page 7:
- 223: Please include a reference after 'in vivo'. Does it matter whether smoke impairs CFTR function if it is already impaired in CF due to mutation? Consider rewording if this is not what was meant here.
Response: Reference added. We have reworded the sentence to emphasize that the impairment was observe in patients without CF.
- 229: It seems odd to lump Australian and global statistics together, then list other individual countries afterwards.
Response: It is odd. We have removed the Australian statistics.
l.238-251 Given the Phe508del is the most common mutation, does it pay to mention which class it belongs to?
Response: Thank you for the comment. We have added that F508Del is a Class II mutation early on in the paragraph, line 257.
Page 8:
- 268: Consider moving this paragraph to the start of the next paragraph and also merging the paragraph starting at l. 276, so that everything on TB is in one paragraph.
Response: This has been merged.
- 276: Do the authors mean carriers? I guess this would be less well studied than CF patients. Please clarify.
Response: Deficiency of arylsufatase B is seen in patients with CF. Not well studied in carriers.
- 277: enzyme that 'is', not 'are'
Response: This has been corrected.
- 282: This sentence seems to not support the argument for why carriers have persisted in the population?
Response: That does not. Thank you for pointing this out, this should have been placed more appropriately. We have moved this to line 307-308.
- 302: Remove full stop after 'indeed'.
Response: This has been corrected.
Page 9:
- 309: Include the recent clinical trial: DOI: 10.1016/S2213-2600(22)00165-5
Response: This has been added.
Page 10:
- 348: Explain why the single and dual therapies weren't suitable for a substantial proportion of CF mutations.
Response: We have included the gene mutations that were eligible for single/dual therapies.
- 350: What type of modulator is elexacaftor?
Response: It is a CFTR corrector, increasing the chloride transport by increase the quantity of functional CFTR delivered to the cell surface. We have added this to the first mention of elexacaftor.
- 351: What do the authors mean by minimal function cohort?
Response: The minimal function cohort refers to no CFTR protein produced or a CFTR protein that does not respond to ezacaftor, ivacaftor, and tezacaftor/ivacaftor in vitro. We have added that to the sentence.
- 371: Can the authors briefly comment on limitations around price, access, funding and approval in different countries?
Response: Paragraph added, lines 397-400.
Page 11:
- 392: How many of total CF patients are affected by Class I?
Response: Around 20%, this has been added and referenced. Line: 421
- 401: Again, I am confused why CFTR modulator therapy is classified as gene therapy.
Response: This has been corrected to CFTR modulator therapy.
- 394: Also in this section, can the authors speculate on the future advances in CF care using gene therapy, i.e. introducing a copy of the functional gene?
Response: Thank you for this comment. Gene therapy is a hope for the future although there are many barriers. Added 2 sentences on gene therapy in CF. Line 437-440.
- 425: Add a hyphen between 'value' and 'adding'.
Response: This has been added.
Other suggestions for improvement:
A lot of what is discussed in the review and the benefit of various modulator therapies relies on understanding the different classes of CFTR mutations. Consider having a table with the different classes, their functional impairment category, examples of mutations, frequency of occurance and effective treatment regime.
Response: Thank you for this idea. We have added table 1 to incorporate all of the suggestions on page 8.
Another Figure could show a scheme of the main points that characterized each chapter of the CF history/treatment/care given that this is the focus of the review as opposed to numerous other reviews that show similar Figures as that in Figure 2.
Response: We are grateful for your comment. We feel the main points are better suited for the narrative as presented in text instead of a schematic timeline.
Reviewer 3 Report
I found this review article a rather refreshing overview on cystic fibrosis etiology, pathogenesis and treatment. I believe it will make a good read for both clinicians and academics interested in the subject. Therefore, apart from the few minor corrections listed below, I recommend the manuscript for publication.
Minor corrections:
Line 233: “unique regulatory T domain” – regulatory R domain;
Line 278: “related the sulfation” – related to the sulfation:
Line 336: “in those aged and above” – in those aged 12 years and above;
Line 402: “treatment throughout pregnant” - treatment throughout pregnancy;
Author Response
Response: Thank you for reviewing our paper and we are grateful for your comments. All these corrections have been made.